# Multimorbidity and its effect on perceived burden, capacity and the ability to self-manage in a low-income rural primary care population: A qualitative study

Ruth Hardman [1,2]*, Stephen Begg[1], Evelien Spelten[1]

**1** School of Rural Health, La Trobe University, Bendigo, Victoria, Australia, **2** Sunraysia Community Health Services, Mildura, Victoria, Australia

* r.hardman@latrobe.edu.au

## Abstract

### Introduction

Multimorbidity is increasing in prevalence, especially in low-income settings. Despite this, chronic conditions are often managed in isolation, potentially leading to burden-capacity imbalance and reduced treatment adherence. We aimed to explore, in a low-income population with common comorbidities, how the specific demands of multimorbidity affect burden and capacity as defined by the Cumulative Complexity Model.

### Materials and methods

Qualitative interviews with thirteen rural community health centre patients in Victoria, Australia. Participants were aged between 47–72 years and reported 3–10 chronic conditions. We asked about perceived capacity and burden in managing health. The Theory of Patient Capacity was used to analyse capacity and Normalisation Process Theory to analyse burden. All data specifically associated with the experience of multimorbidity was extracted from each burden and capacity domain.

### Results

The capacity domains of biography, resource mobilisation and work realisation were important in relation to multimorbidity. Conditions causing functional impairment (e.g. chronic pain, depression) interacted with physical, psychological and financial capacity, leading to biographical disruption and an inability to realise treatment and life work. Despite this, few people had a treatment plan for these conditions. Participants reported that multimorbidity affected all burden domains. Coherence and appraisal were especially challenging due to condition interactions, with clinicians providing little guidance.

### Discussion

The capacity and burden deficits highlighted by participants were not associated with any specific diagnosis, but were due to condition interactions, coupled with the lack of health

**Data Availability Statement:** Restrictions have been placed on our dataset by the La Trobe University Human Ethics board. Data cannot be publicly shared because it consists of qualitative

interviews with patients conducted in two rural communities where there may be risk of identification. Data are available through the La Trobe University ethics committee or from the corresponding author for researchers who meet the criteria for access to confidential data. Contact information for the La Trobe University Human Ethics board: humanethics@latrobe.edu.au or phone +61 3 9479 1443. The approval reference number is HEC19387.

**Funding:** The authors received no specific funding for this work.

**Competing interests:** The authors have declared that no competing interests exist.

provider support to navigate interactions. Physical, psychological and financial capacities were inseparable, but rarely addressed or understood holistically. Understanding and managing condition and treatment interactions was a key burden task for patients but was often difficult, isolating and overwhelming. This suggests that clinicians should become more aware of linkages between conditions, and include generic, synergistic or cross-disciplinary approaches, to build capacity, reduce burden and encourage integrated chronic condition management.

## Introduction

The shift from acute to chronic health conditions as the main driver for worldwide burden of disease has demanded alternative healthcare solutions [1]. More recently, there has been a recognition that many chronic conditions do not exist in isolation, but as clusters of conditions [2]. Multimorbidity, which is defined as the presence of two or more chronic health conditions [3], has become the rule rather than the exception [2,4], especially with increasing age. In common with individual chronic conditions, multimorbidity is also more prevalent in vulnerable groups, including rural [5] and socially deprived populations [4].

This 'new normal' of multimorbidity is not reflected in our health systems, models of care or everyday clinical practice. Although the development of the Chronic Care Model [6] has enabled many healthcare systems and practitioners to transition from acute to chronic care, it remains limited by its single disease focus. Studies of clinical guidelines and qualitative studies with patients and healthcare providers (HCPs) note that multimorbidity is difficult to integrate into a chronic care model due to conflicting treatment recommendations, condition interactions and excessive treatment burden [7–10].

Traditionally, multimorbidity has been understood as a list of separate conditions which are prioritised according to mortality risk [11,12]. In clinical practice, this has led to each condition being managed as a separate entity [10], with precedence given to conditions with a higher risk of future adverse outcomes such as diabetes or cardiovascular disease [13,14]. Interviews with patients suggest that they approach multimorbidity differently, placing greater importance on symptomatic conditions affecting their quality of life [15–19]. This preference has implications for health outcomes, with conditions that may have low symptom burden but high future risk being deprioritised or ignored by patients [16].

Recognising that for most people, multimorbidity is an experience they live with, rather than a condition(s) they die from, researchers have started to pay more attention to the patient experience [16,20–22]. This has drawn out the importance of interactions between the disease (s) and psychosocial factors. The risk of a co-occurring mental health condition (often excluded from morbidity counts) [11] increases with each additional physical condition [23], and socially disadvantaged populations report 10–15 years earlier onset of multimorbidity [4]. Although disease count is important when measuring mortality, functional impairment, psychological distress and social context are more accurate predictors of quality of life [11,24].

In acknowledgement of these social and contextual influences, Coventry [21] has characterised multimorbidity as an 'encounter with complexity', consisting of emotional, environmental and functional as well as medical components. Shippee's Cumulative Complexity Model [25], which defines complexity as the result of an imbalance between an individual's capacity and their workload, is a useful way to understand multimorbidity. This model conceptualises capacity as a persons' physical, cognitive and psychological functioning as well as their

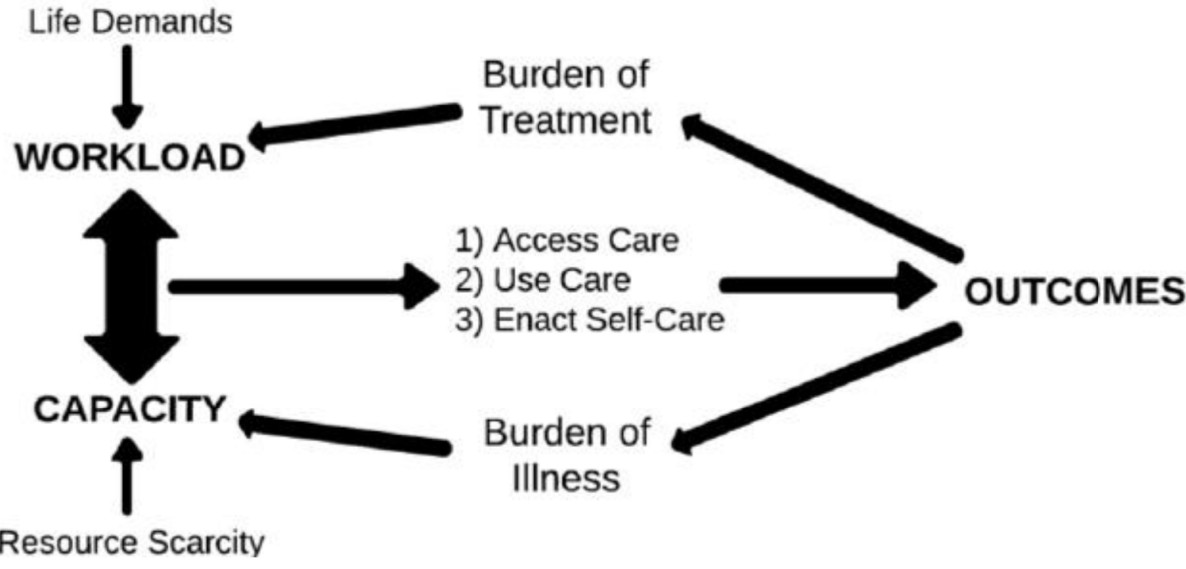

**Fig 1. The cumulative complexity model [25,28].**

available resources. Workload comprises treatment and condition requirements plus the demands of everyday life (see Fig 1). Although Shippee's model [25] uses the term 'workload', to increase clarity and consistency with the wider literature we will be using the term 'burden' or 'treatment burden' instead, defined as both the healthcare tasks ('work') of managing chronic illness, and the impact on the patients' life roles and functioning ('life') [26].

Concepts of burden and capacity are important in multimorbidity, since the additional treatment tasks (e.g. medications, condition monitoring, appointments) associated with multiple conditions are likely to increase treatment burden. With sufficient capacity, the burden can be managed; but low capacity (e.g. inadequate income or social support) will reduce a persons' ability to manage their treatment burden (e.g. medication costs, accessing appointments). Burden-capacity imbalance can lead to reduced treatment adherence and declining health outcomes [27]. This model is particularly relevant to socially disadvantaged populations, because they experience higher levels of multimorbidity [4] (therefore greater treatment burden) whilst having fewer resources (lower capacity).

The cumulative complexity model has been explored in a range of populations, including people with diabetes [29], kidney disease [30], stroke [31] and in low-middle income countries [32]. We wished to apply this model to a rural low-income multimorbid population, who were at risk of both high burden (from multiple health conditions) and low capacity (from resource constraints). The point of difference in this study was its focus on how the experiences that are specific to multimorbidity affect perceptions of burden and capacity.

To explore this we will use established taxonomies of workload and capacity, since this will enable us to see how each workload or capacity domain is differentially affected by the demands of multimorbidity. The Theory of Patient Capacity [33], which describes capacity as the interaction between Biographical adjustment, Resource mobilisation, Environmental fit, Work realisation and Social functioning (abbreviated as 'BREWS') will structure our examination of capacity. To explore burden, we will use Normalisation Process Theory (NPT). This theory explains how new practices are integrated into everyday life [34], and has been applied previously in studies of treatment burden [27,31,35].

Our research question was: In low-income rural primary care patients, how does the experience of multimorbidity affect perceived burden and capacity to self-manage their health?

## Materials and methods

### Study design

The study method was qualitative. We employed a phenomenological methodology, which is an approach focussed on the lived experience of participants [36]. Research was conducted in accordance with national ethics guidelines, with approval granted by the La Trobe University Human Research Ethics Committee (HEC19387). The completed COREQ checklist for reporting of qualitative studies is available in S1 File.

### Participant recruitment and setting

Participants were clients of two regional community health centres in Victoria, Australia. Victorian community health centres provide primary care and chronic disease services to low-income and socially disadvantaged populations [37]. People aged between 18–75 years who described themselves as having at least two chronic physical health conditions, such as diabetes, back pain, arthritis, heart or lung conditions were invited to participate. Our focus was on conditions commonly managed in primary care. Since low-income groups are known to experience multimorbidity 10–15 years earlier [4], we looked for people who were middle-aged or early retirees (under 75 years). We were interested in exploring multimorbidity in an age group where there are still societal expectations of active and independent life roles.

Participants were recruited via posters in the waiting rooms of the health centres, as well as by direct invitation from their healthcare providers. Potential participants were provided with basic study information and their contact details were provided (with permission) to the researchers. Sixteen people expressed interest in the study, with three withdrawing prior to the interview. Recruitment was initially via snowball sampling, with the last four participants purposively selected to ensure gender balance.

### Data collection

Following completion of written consent, we conducted semi-structured interviews, each lasting for approximately one hour. All interviews were conducted by a single clinician-researcher (RH), either by phone or at a community health centre. The interview protocol was developed following review of the qualitative literature [31,33,38,39], but was not trialled in patients. Interview topics explored all capacity and burden domains as outlined by BREWS and NPT. We asked people to describe their health conditions; how their daily life was affected; the treatments they needed to undertake and the difficulties they experienced in managing their healthcare. Interviews were audio recorded and continued until all researchers agreed that saturation had been reached. Interviews were transcribed verbatim by the interviewer (RH). Field notes detailing key issues and observations were made following each interview. The interview protocol is available in S2 File.

Participants also completed a series of self-report scales and sociodemographic details were recorded.

### Data analysis

We aimed to explore how the experience of multimorbidity, as distinct from that of having a single chronic condition, affected each aspect of capacity and burden. For this reason, we undertook analysis in several stages (Fig 2). First, we explored capacity and burden by dividing

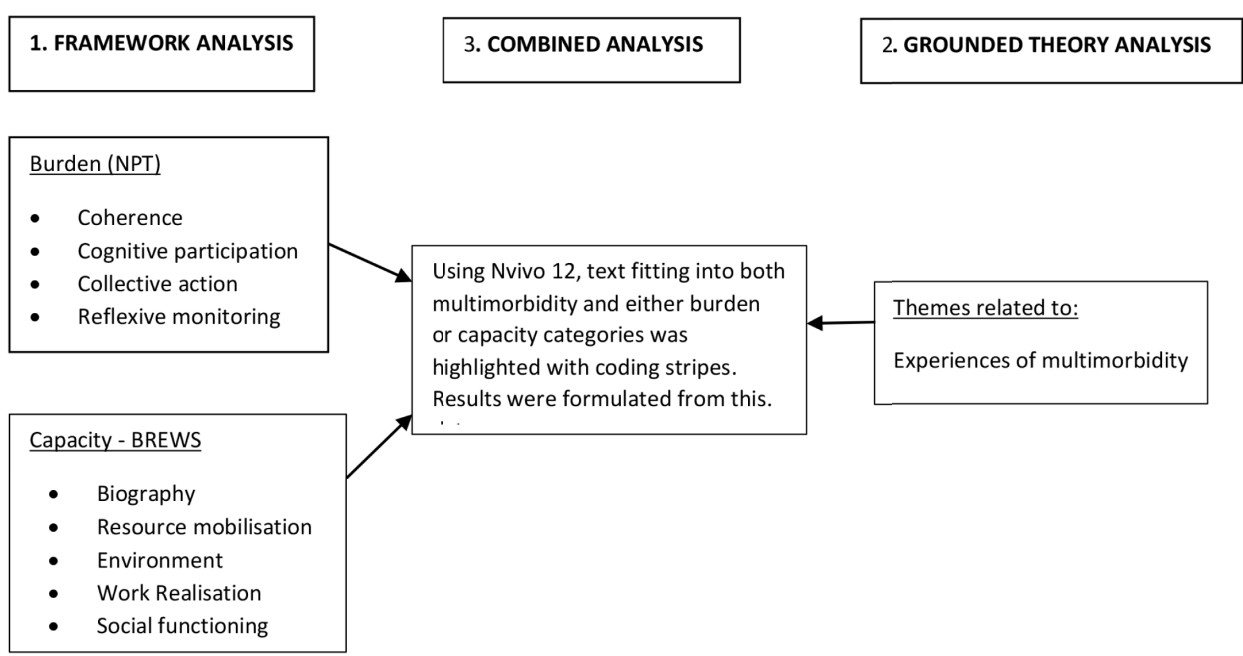

**Fig 2. Description of analysis process.** All data underwent initial framework analysis using the two categories of NPT and BREWS. We then returned to the raw data to record experiences of multimorbidity. Finally, analyses were combined to identify multimorbidity data that was relevant either to burden or capacity.

the interview data into these two broad categories. We then applied framework analysis, which uses a pre-defined coding system (the framework) to structure the data [40]. The coding systems used were the Theory of Patient Capacity to analyse capacity [33] and Normalisation Process Theory to analyse burden [27]. Tables 1 and 2 describe each coding system in terms of its component domains; further details are available in S3 File. Data was transcribed verbatim by RH and initially coded by hand, then imported into NVivo 12. Coding was evaluated and refined by SB and ES. Findings were reviewed and disagreements resolved in discussion with all three researchers.

Since our focus was on the relationship between capacity, burden and the experience of multimorbidity, we then returned to the original data and performed a second analysis, using a grounded theory approach to identify themes related to multimorbidity. Grounded theory is an inductive approach to qualitative research that focusses on the data alone, without an underlying theoretical perspective [36]. We looked for any references to having more than one health condition, including how conditions were prioritised, interactions between conditions and any demands related to managing multiple health conditions. Following the second analysis, by using the coding stripes function on NVivo 12, we could then locate all data associated with both the burden/capacity and the multimorbidity codes. Thus, we could identify the burden and capacity domains perceived by the participants to relate most strongly to multimorbidity.

## Results

### Participant and interview characteristics

Eleven interviews were conducted with thirteen people (two interviews were with couples who both experienced multimorbidity). Nine interviews were conducted face-to-face at a

**Table 1. Coding domains for capacity (BREWS).**

| CAPACITY DOMAINS | | |
|---|---|---|
| **Biography** | | Ability to maintain purpose and create a meaningful life while living with chronic conditions |
| **Resource mobilisation** | Physical | Symptom burden (pain, fatigue etc.), functional capacity (task performance, physical fitness, sensory abilities). |
| | Psychological | Personal traits (resilience, self-efficacy); mental health burden (anxiety, depression); cognitive capacity (memory, literacy). |
| | Practical | Financial, personal (e.g. access to transport) and organisational (e.g. aids/equipment, governmental services) resources. |
| **Environment** | | Support available in healthcare and personal environments; whether treatment demands are a good 'fit' with daily life. |
| **Work realisation** | | Ability to successfully achieve and normalise all aspects of treatment workload; ability to achieve expected life roles. |
| **Social functioning** | | Ability to socialise; practical social supports, social acceptance or stigma, social relationships with HCPs. |

community health centre, and two by phone, due to COVID-19 restrictions. Interview duration ranged from 31–71 minutes (mean 43 minutes). Participants were aged between 47 and 72 years (mean = 61 years) and reported between 3–10 health conditions each (mean = 7) using the Disease Burden Impact Scale [24,41] to report type and severity of condition. The most common conditions reported were musculoskeletal disorders (back pain, osteoarthritis and/or other chronic pain—reported by 100% of participants), followed by type 2 diabetes (n = 10 people, 77%); cardiovascular conditions (heart disease, peripheral vascular disease and/or hypertension: n = 10, 77%); overweight/obesity (n = 8, 62%); mental health conditions (depression, anxiety and/or PTSD: n = 8, 62%) and gut or bowel disorder (n = 8, 62%). Other conditions reported by 3–6 participants were respiratory conditions (asthma and COPD), vision and hearing impairments. Table 3 records key characteristics of the participants.

## Multimorbidity and capacity

As illustrated in Table 4, multimorbidity was related to biography, resource mobilisation and work realisation. People reported biographical challenges when a new condition emerged. They had often managed a chronic condition for years without difficulty, but the impact of another condition could make all the difference. This was especially the case with conditions associated with functional impairment, which often placed greater demands on biographical reframing due to the loss of meaningful activities (especially if people had to stop work or lost other significant life roles).

**Table 2. Coding domains for burden (NPT).**

| BURDEN DOMAINS | |
|---|---|
| **Coherence (Sense-Making)** | Learning about, understanding and making sense of the condition(s) and treatments, planning care, setting goals. |
| **Cognitive Participation (Relationship work)** | Engaging with others (HCPs, services, friends) for help, managing these relationships; individual organisational tasks to support healthcare (e.g. transport, arranging prescriptions). |
| **Collective Action (Enacting work)** | Specific treatment tasks (appointments, medication, self-care); integration of condition and treatment into daily life (adjusting to work, social or financial changes). |
| **Reflexive Monitoring (Appraisal)** | Reflecting on the condition(s) and treatment, reviewing and modifying management individually or in discussion with others. |

**Table 3. Characteristics of study participants.**

| ID | Sex | Age | Living situation | Source of income | Health conditions |
|---|---|---|---|---|---|
| P1 | M | 57 | With friend | Unemployment payment[1] | Back pain, OA, other chronic pain, depression, PTSD, liver disease, vision. |
| P2 | F | 50 | Spouse and child | Unemployment payment[1] | T2DM, back pain, other chronic pain, obesity, depression, gut, bowel, vision, HT |
| P3 | M | 72 | Spouse (P4) | Age pension[2] | RA, back pain, OA, CVD, HT, gut, vision, overweight |
| P4 | F | 71 | Spouse (P3) | Age pension[2] | RA, T2DM, back pain, OA, overweight, gut, bowel, asthma |
| P5 | M | 70 | Spouse | Age pension[2] | CVD, HT, T2DM, PVD, vision, hearing, OA, kidney disease |
| P6 | M | 54 | Alone | Unemployment payment[1] | T2DM, OA, back pain, other chronic pain, PVD, HT, overweight, vision, depression, thyroid. |
| P7 | M | 65 | Spouse, other family | Part time work[3] | T2DM, HT, back pain, other chronic pain, gut, depression/anxiety, sleep apnoea, obesity, hearing |
| P8 | M | 59 | Alone | Unemployment payment[1] | T2DM, PVD, overweight, depression/anxiety, OA, back pain, other chronic pain. |
| P9 | F | 57 | Children | Disability pension[2] | T2DM, OA, back pain, gut, COPD, asthma, depression/anxiety, incontinence, HT |
| P10 | F | 66 | Spouse (P11) | Part time work[3] | OA, asthma, depression/anxiety |
| P11 | M | 68 | Spouse (P10) | Age pension[2] | CVD, HT, T2DM, PVD, hearing, cancer, gut, asthma, depression/anxiety, COPD, chronic back pain, other chronic pain |
| P12 | F | 47 | Other family | Carer pension[2] | T2DM, OA, other chronic pain, back pain, kidney disease, liver disease, cancer, obesity, gut, bowel, HT |
| P13 | F | 60 | Alone | Disability pension[2] | T2DM, OA, back pain, other chronic pain, HT, obesity, COPD, gut, lymphoedema, sleep apnoea |

CVD = cardiovascular disease; HT = hypertension; T2DM = type 2 diabetes; COPD = pulmonary disease; RA = rheumatoid arthritis; PVD = peripheral vascular disease; OA = osteoarthritis; PTSD = post-traumatic stress disorder.

1 = income ≈ A\$15000 p/a–below poverty line; 2 = income ≈ A\$22000 p/a–equivalent to Australian poverty line; 3 = unskilled occupation, < 20hr/week.

*Participant 6: I've always been an outdoor labouring person working all my life you can't just flick the switch and sit in front of a computer I'd rather shoot myself to be honest [I feel] just not as happy. . .because you're not going forward. . .in life because you haven't got a job. . .it's like you're just stagnant*

For those people who were waiting (or hoping) for a definitive diagnosis, treatment or explanation of their condition, building biographical capacity was difficult. They felt that they were in limbo and unable to 'move on' with their lives.

Multimorbidity had a profound effect on resource mobilisation. Physical, psychological and financial capacity were all compromised. Eight of the thirteen participants reported chronic pain conditions (osteoarthritis (3), shoulder pain (1), back pain (2), leg/foot pain (2)) and two reported diabetic foot ulcers as their most important condition. All ten participants related condition priority to the associated loss of physical capacity, including the ability to

**Table 4. The relationship of multimorbidity to capacity domains.**

| CAPACITY DOMAINS | | THEMES RELATED TO BOTH CAPACITY AND MULTIMORBIDITY |
|---|---|---|
| **Biography** | | Each new condition requires biography work. Certain conditions (e.g. undiagnosed, disabling) place greater demands on biography. |
| **Resources** | Physical | Conditions causing functional impairment are prioritised. |
| | Psychological | Poor mental health affects ability to look after other conditions. |
| | Financial | Multiplying healthcare costs. 'Tipping point' where increased number of conditions or disability results in loss of income. |
| **Environment** | | No issues specific to multimorbidity |
| **Work** | | Treatment workload is easier to achieve if conditions have low symptom burden or are perceived as interrelated; harder if mental health is poor. |
| **Social** | | No issues specific to multimorbidity |

work, exercise, undertake household tasks and leisure activities. Both couples prioritised their partner's chronic pain condition above their own chronic pain conditions because of the additional physical demands it placed on them as carers, further reducing their own (already restricted) physical capacity.

> *Participant 11*: *[my biggest issue is] the shoulders more than diabetes. . .because if I do something I shouldn't do I pay for it. . .Participant 10 (spouse): and it impacts on sleep and me having to do things*

Psychological capacity was also essential. Although we selected participants based on their physical health conditions, eight people also reported a mental health diagnosis. The remaining five interviewees also described emotional difficulties, with several having undergone mental health treatment in the absence of a formal diagnosis. Two participants rated depression as their most important condition, and one prioritised obesity due to its impact on her mental health. Again, these conditions were prioritised because they prevented the attainment of desired goals including the ability to socialise, work, undertake study, and engage in family life. Participants also described how depression affected their adherence to, and motivation for, treatment of other health conditions. All those having trouble with their diabetes management reported moderate to severe depression.

> *Participant 1*: *I get depressed because things don't seem to happen quickly enough for me and I get upset that I can't do things so I don't eat, I stop taking my meds, I self-harm . . . things like that*

All participants noted that multiple chronic conditions led to increased healthcare expenditure, thus reducing financial capacity. All bar one interviewee stated that they had not undertaken recommended treatments or appointments at times due to cost.

> *Participant 12*: *the psychologist that I'm seeing I. . . pay out-of-pocket to see her. . .I have to think about what don't I get done that week do I not pay my phone or power. . .*

Four participants paid for private health insurance. Although there was a recognition that this provided a better quality, faster service, participants felt that the cost could not be sustained into the future without additional funds provided by ongoing employment or other family members.

Increased healthcare costs were often complicated by loss of income. As multimorbidity increased, functional capacity declined, with ten of the thirteen participants reporting that their health conditions had forced them to stop work. Several people described a 'tipping point' where they were no longer able to work due either to a gradual increase in disability or due to a new health condition which resulted in greater functional impairment. Most were unable to access the disability or aged pension (at least initially), which could provide a low but secure income, and were reliant on savings or financial support from their family.

> *Participant 6*: *I've lost my house that was the main thing. . .I nearly had it paid off [but] I had no insurance because I had shoulder operations before and they wouldn't give me income insurance so I couldn't get that . . .when this happened I was buggered couldn't work so I had to sell my house*

The ability to build capacity by work realisation depended on the nature of the condition. Conditions such as diabetes which had a low symptom burden, were reasonably predictable and had a clear management plan were cited as easier to successfully manage than more unpredictable or difficult to control conditions such as chronic pain, rheumatoid arthritis or depression.

> *Participant 4*: *[managing condition workload] it depends on what sort of health conditions you've got because my diabetes is really just diet and of course medication but the rheumatoid arthritis is one that you need to keep in check . . .if you have a flare-up*

If the person saw their conditions as interrelated in terms of cause or treatment, they were more able to manage it, compared to seeing it as a series of separate conditions. Those with the greatest difficulty in successfully accomplishing treatment work all reported mental health issues, associated with a sense of being overwhelmed and disorganised, rather than enormous treatment demands.

> *Participant 2*:*. . . it'd be so much easier if I just had one health problem I could work on and not have multiple problems and you just think . . .put your hands up. . .I got really bad a few months ago . . ..I just stopped taking everything . . ..I went into a really deep depression and couldn't be bothered doing a thing*

**Interacting capacities.** Loss of capacity often snowballed. The interaction between mental and physical health conditions was a common theme. Some thought of depression as the trigger for all their health conditions, often related to past trauma. For others, depression developed after other health conditions, either directly (e.g. following heart surgery) or due to pain or functional incapacity.

> *Participant 8*: *that's where depression comes in you're just sitting in the same house all day every day when I was working I would have holidays for 8–10 weeks a year*

Loss of physical capacity, in turn, provided multiple triggers for mental health decline. It had direct impacts on income (ability to work) and on biography (loss of life role, ability to engage in meaningful activity), as well as the symptom burden of pain or fatigue. Worsening depression, whether triggered by a physical health issue or not, affected work realisation, reducing adherence to treatment tasks, affecting motivation and problem-solving ability. It could thus exacerbate co-existing physical health conditions.

Financial resources could bolster capacity. Those who had a secure (if limited) income, compared to those receiving unemployment benefit or in an insecure work environment had fewer mental health difficulties and more effective strategies to manage their mental health. The two participants working part-time chose to continue because they recognised the mental health benefits (boosting psychological capacity) despite the fact that it exacerbated their chronic pain (reducing physical capacity).

> *Participant 7*: *with depression people handle it in different ways I keep busy I work I do things if I can't work what happens I go downhill. . .as soon as I stop doing things I go downhill*

All three resource mobilisation factors were closely related to biographical disruption (Fig 3). Reduced physical capacity led to the loss of preferred and meaningful activities, including

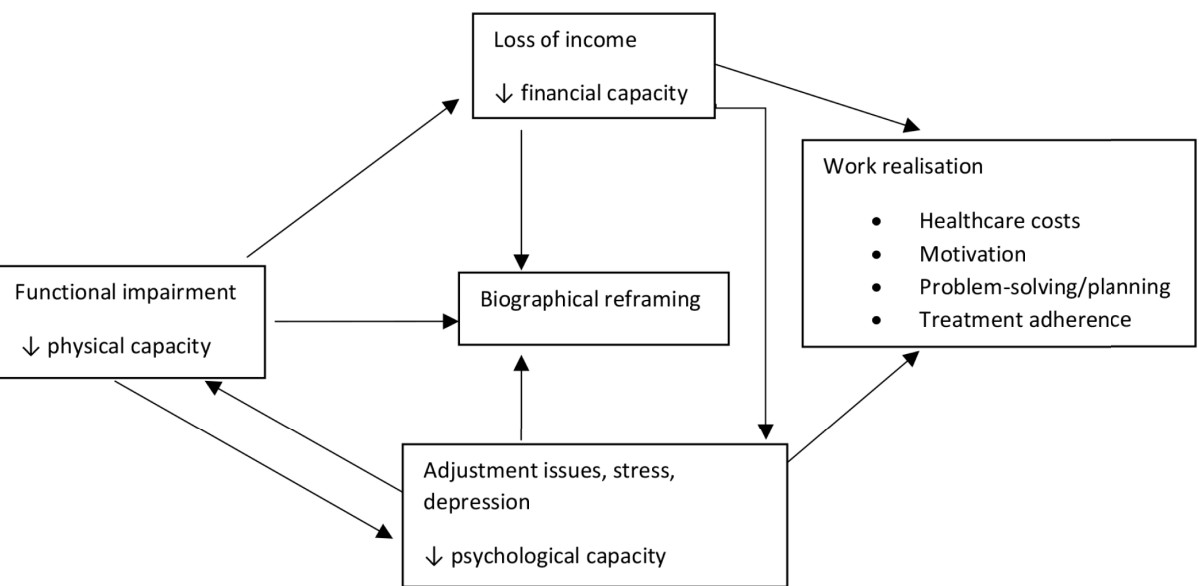

**Fig 3. Interacting capacities.** Functional impairment leads to loss of income, biographical difficulties and psychological stress. Loss of income affects biography, psychology and work realisation. Psychological stress affects biography, physical capacity and work realisation.

important life roles such as work. This could lead to depression (reduced psychological capacity), which then affected motivation and future planning. Reduced financial capacity often resulted from the loss of physical capacity (inability to earn an income) but lack of income also limited people's access to meaningful or enjoyable activities, as well as access to healthcare (which could potentially improve physical and psychological capacity). Those with greater financial security (e.g. access to the pension) were more able to put their energies into meaningful activity which assisted with biographical reframing.

## Multimorbidity and burden

The relationship between multimorbidity and the different aspects of burden, as described by NPT, is demonstrated in Table 5.

The ability to form a <u>coherent</u> understanding of health problems was easier if the conditions were seen to be interconnected or to stem from the same cause. Disparate health issues often felt overwhelming and some people struggled to make sense of them. These participants often had depression as their primary (initial) health condition.

**Table 5. The relationship of multimorbidity to burden domains.**

| BURDEN DOMAINS | | THEMES RELATED TO BOTH BURDEN AND MULTIMORBIDITY |
|---|---|---|
| Coherence | | Making sense of conditions is easier when they are interconnected but harder if depression dominates. HCPs help with diabetes understanding but less so with other conditions. |
| Cognitive participation | HCP relationships | Multiple HCP involvement, poor service co-ordination between conditions |
| | Individual | Mental health affects ability to organise healthcare |
| Collective action | Treatment tasks | More tasks to undertake (polypharmacy, appointments, self-care), but for many this becomes a routine not a burden. |
| | Contextual Integration | Greater healthcare costs, often combined with loss of income, are the main barrier |
| Reflexive monitoring (Appraisal) | | Constant need to reassess due to interactions between conditions and treatments. Little guidance or assistance from HCPs. |

*Participant 5 [managing multiple health conditions is not a problem]. . .because I got them all together and they're sort of all related. . .I was diagnosed [with diabetes] in 89 and I had my first heart attack in 92 so it's the same period of time and there is no doubt about the fact that they were all related to my drinking. . .I see it as one big problem instead of separate things*

Many people reported that HCPs had provided them with information and education about diabetes, but few other conditions were addressed. Of note, HCPs were rarely seen as sources of information about conditions causing functional impairment (pain, mental health, fatigue). Learning about these conditions was either via trial and error or the internet.

*Participant 10 . . .[learning about osteoarthritis] I've self-managed I've experimented with myself*

In the cognitive participation domain, interactions with HCPs multiplied as numbers of health conditions increased. All participants reported involvement with several providers, with most seeing 3–5 HCPs regularly. This could be challenging if HCPs were time-limited, unsympathetic or transient (common in rural areas). Some people found it hard to keep track of who they were seeing and for what condition. Multidisciplinary and co-located services were noted to be very helpful and the need for co-ordination was repeatedly discussed.

*Participant 11: every time you go there [to the GP clinic] they change the doctors around and the doctor changes the tablets Interviewer: seeing different doctors all the time? Participant 11: yeah*

Individual organisation was important in multimorbidity. Those who had a routine, or a system to manage medication coped better. Poor medication adherence was associated with a lack of routine and was frequently associated with mental health conditions.

The topic of collective action was explored in relation to the numbers of treatment tasks, and the ability to integrate the condition into daily life. All participants were asked which of their health condition(s) required the most treatment work. Four people selected diabetes as having the greatest workload and three nominated conditions related to wound healing and dressing. The remaining participants did not identify a specific condition. Six of those with diabetes felt that their diabetes management was normalised and fully integrated into their lives.

Despite this, all participants recognised that having multiple health conditions meant additional treatment tasks. Some incorporated it into their daily routine, with several people describing it as their 'job', but others found the workload too great. Everyone reported polypharmacy, and many had concerns about medication interactions and side-effects. Those living more remotely (6 hours travel from the state capital city) had significantly greater time and money costs associated with travelling to appointments, as well as fewer treatment choices, compared to those living in an inner regional area (2 hours from the capital). Some people managed their treatment load by recognising that the same treatment (e.g. exercise) could work for several conditions.

For most people, the additional costs of healthcare associated with multimorbidity was the main barrier to the integration of the conditions into their daily life, especially for those who were no longer able to work.

Finally, reflexive monitoring (appraisal) played an important role in the management of multimorbidity. People had to undertake more cognitive work to understand how treatments and conditions interacted, and needed to constantly reassess and reconsider one condition in

the light of their other conditions. This ongoing instability could make normalising treatment workload more difficult. Participants reported undertaking appraisal in relation to both medication use and lifestyle recommendations. Most people had concerns about polypharmacy and were keen to minimise medication use, but struggled to unravel the interactions between conditions and medications.

> *Participant13*: *because it's all combined as I said when I went for oncology there yesterday because I take a tablet that can cause hot flushes. . .they ask do you get hot flushes. . . I don't know I've got so many health problems. . .*

Lifestyle recommendations were often questioned because participants felt they were unrealistic (for diet), or because people did not know how to undertake exercise when they had coexisting chronic pain.

> *Participant 3*: *everywhere we go it whether you go to see the GP or [the dietician] . . .the physiotherapist or whatever they all say exercise and I said but it's just not possible we can't do it. . .because of the pain*

Those with diabetes frequently described the process of appraising and modifying treatment due to the impact of stress, pain or illness on their blood sugar levels. Some were confident in 'trouble shooting' their various health conditions and could monitor and adjust treatment as needed, while others found that additional health conditions 'muddied the waters' and made it harder to plan what to do.

> *Participant 7 [managing diabetes when first diagnosed] because it was new it was a bit of a novelty and I knew what I had to do but as time goes on. . .I've had lots of other health issues. . .I have to think oh I've got to look after my shoulder I've got to be careful of my hernia and it takes you away from the diabetes*

Many participants engaged in individual appraisal and adjusted their treatments (including medication) without necessarily discussing the changes with a HCP.

> *Participant 8*: *when I was going to [the hospital] they wanted four [blood sugar] readings a day but you run out of the strips after a while. . . I did that for about 4 or 5 weeks but it there wasn't really a great deal gained by it so I can't see the point.*

## Discussion

### Main findings

This study aimed to investigate how the additional challenges of multimorbidity influence different aspects of capacity and burden, as described in the literature. For this rural, low-income population, the nature of the condition was of key importance. Conditions associated with functional impairment, especially chronic pain and mental health conditions, had the greatest influence on capacity. In our analysis of burden, multimorbidity was associated with a greater number of treatment tasks, costs, and appointments with HCPs, as has been well-documented previously [42,43]. The domains of coherence (sense-making) and reflexive monitoring (appraisal) were particularly important and this was related to the interactions between capacities, conditions and treatments that most participants dealt with.

## The nature of the condition

Several studies [15,16,18,22], have reported that patients prioritise health conditions based on their functional impact, and this study reports similar findings. Analysis of specific capacity domains showed that pain and mental health conditions (as well as diabetic foot ulcers) had the greatest impact on capacity. In this low-income rural setting, most participants had been manual workers, and the biggest functional impact was the loss of employment. This had multiple impacts on other capacity domains including biographical, financial, psychological and work realisation, and could affect motivation for, and adherence to, self-management of other health conditions [21,39].

The loss of capacity associated with functional and psychosocial conditions indicates the need for targeted treatment to bolster capacity. However, for many participants, treatment work was focussed on conditions with few symptoms (e.g. diabetes), with limited formal treatment for their chronic pain or depression. In an ideal world, per the Cumulative Complexity Model, successfully managing treatment work should reduce illness burden and increase capacity, thus making it easier to normalise health conditions. In this population, effectively managing treatment work often had little impact on capacity, since loss of capacity was related to conditions which had few treatment demands. Without observed capacity benefits, this may discourage people from engaging in treatment work [44]. Although the Cumulative Complexity Model and the associated burden and capacity frameworks fitted the data well, the issue of mismatch between treatment burden and capacity deficits has not been previously noted. This may be an important factor in multimorbidity self-management which deserves further attention.

While some participants had structured management approaches for their pain or mental health conditions, this was often developed without HCP input. Others did not see such conditions as having a treatment pathway at all, but just as symptoms to endure. However, these conditions are often responsive to generic interventions such as exercise or mindfulness, which means that their management need not increase treatment work: the use of synergistic treatments which work across a range of health conditions has been recommended for multimorbidity [10,19,22]. The challenge may lie in convincing patients of treatment efficacy. Despite its known efficacy for chronic pain and depression [45], many participants believed that exercise was contra-indicated, or did not know how to approach it. This may be an important but neglected role for HCPs working in chronic disease management. Providing education about the relationships between pain, mental health and other chronic conditions, as well as synergistic treatments such as exercise and mindfulness could be helpful, although a low HCP knowledge base in these areas [46,47] and insufficient funding of non-pharmacological interventions (noted by several participants) remains a barrier.

The role of mental health in treatment adherence makes it a particularly important area to be formally addressed in the chronic disease management environment. For all participants, their mental health was closely entwined with, and responsive to, their physical and financial capacity. Unfortunately, as a relic of dualism, mental health conditions are often dealt with and funded in isolation from other chronic diseases, and many mental health providers have limited knowledge of physical health conditions and limitations. This study emphasises the importance of ensuring that mental health interventions are integrated and tailored to people with co-existing physical health conditions, thus reflecting the reality of people's experience as unified beings, not as minds and bodies.

## Interactions and integration

The increase in treatment tasks and HCP interactions as a result of multimorbidity is widely recognised, and has led to the development of treatment burden assessment tools [26,48,49].

There has been less attention paid to the importance of coherence and appraisal, which emerged as important in this study. Participants frequently engaged in coherence and appraisal work to help them understand and manage the interactions between capacities, conditions and treatments. Psychological, physical and social capacities were inextricably linked, and although stand-alone mental health treatment was important for some people, understanding the connections between, and integrating all three aspects of well-being was the key for those who were managing well.

In terms of individual conditions, people struggled more to make sense of conditions with functional impairment, but this may have been related to the lack of HCP input for these conditions. People reported a greater knowledge of diabetes, with most having received education, but few understood how it interacted with other conditions. Those who saw the linkages between their conditions had a more integrated understanding of their health overall and reported greater confidence in self-management and lower perceived burden.

Although study participants regularly engaged in appraisal, reviewing, prioritising and adjusting their treatments, HCP input into these decisions was limited. Many participants considered that 'juggling' their different conditions was up to them and that the HCPs' role was to provide instruction or treatment on specific individual conditions. An important role for HCPs, which would potentially increase treatment adherence and complement the provision of 'synergistic' treatment interventions, might be to help patients explore the linkages between conditions [20,22]. Making treatment decisions based on a good understanding of how different conditions interact and affect capacity and workload is likely to be useful for both the patient and the HCP.

## Limitations and strengths

This was a small qualitative study of a low-income rural population, and therefore the observations may be less relevant to more advantaged urban groups or in countries with greater levels of social medicine than Australia. The fact that two interviews were conducted by phone (due to Covid-19) could be considered a limitation, although no difference was noted in the interview length or topics covered. The findings remain useful because the research participants came from the most relevant population, as distilled from the literature [4]. Multimorbidity in younger (pre-retirement) age groups is becoming more common especially amongst low-income populations, and there is a need to explore more effective self-management interventions for this group. Our phenomenological focus of prioritising individual experience meant that we could explore a wide and varied range of responses from people facing similar life challenges. The use of existing taxonomies, allowing us to explore different aspects of capacity and burden, was a further strength of this study.

## Conclusion

Our exploration of burden and capacity in this qualitative study confirmed the importance of understanding multimorbidity in its broadest sense. Multimorbidity consists of far more than a list of diagnoses, and to manage multimorbid chronic conditions effectively, HCPs must address the crucial and interacting role of functional and psychosocial factors. Additionally, understanding the links between conditions is important to help patients to integrate and normalise their conditions into their daily life. Patients need support from HCPs to build bridges between conditions and make choices that best fit their needs and preferences. Finally, this study also highlighted the overwhelmingly negative effect of financial insecurity on burden and capacity. Financial hardship associated with chronic illness is well-known [50]. The

additional impact experienced by those who are already disadvantaged underlines the importance of health and social policies to address the challenges faced by this population.

## Supporting information

**S1 File. COREQ checklist.**
(PDF)

**S2 File. Interview protocol.**
(DOCX)

**S3 File. Burden and capacity coding.**
(DOCX)

## Acknowledgments

The authors acknowledge the support provided by staff at Sunraysia Community Health Services and Bendigo Community Health Services, in identifying suitable research participants.

## Author Contributions

**Conceptualization:** Ruth Hardman, Stephen Begg, Evelien Spelten.

**Formal analysis:** Ruth Hardman, Stephen Begg, Evelien Spelten.

**Investigation:** Ruth Hardman.

**Methodology:** Ruth Hardman, Evelien Spelten.

**Project administration:** Ruth Hardman.

**Supervision:** Stephen Begg, Evelien Spelten.

**Validation:** Ruth Hardman, Stephen Begg, Evelien Spelten.

**Writing – original draft:** Ruth Hardman.

**Writing – review & editing:** Stephen Begg, Evelien Spelten.

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
