## [Decision Letter · Decision Letter 0]

27 Apr 2021

PONE-D-20-39711

Multi-morbidity and its effect on perceived burden, capacity and the ability to self-manage in a low-income rural primary care population: a qualitative study.

PLOS ONE

Dear Dr. Hardman,

Thank you for submitting your manuscript to PLOS ONE. After careful consideration, we feel that it has merit but does not fully meet PLOS ONE’s publication criteria as it currently stands. Therefore, we invite you to submit a revised version of the manuscript that addresses the points raised during the review process.

We look forward to receiving your revised manuscript.

Kind regards,

Filipe Prazeres, MD, MSc, Ph.D.

Academic Editor

PLOS ONE

Journal Requirements:

2. In the Methods section, please provide additional information on the interviewers background and training, the number of interviewers and details regarding the length of the interview and where the interview was conducted.

3. We noted in your submission details that a portion of your manuscript may have been presented or published elsewhere.

"Figure one has been taken from another published manuscript. The figure is available through Creative Commons and the authors have been appropriately cited in the manuscript."

Please clarify whether this publication was peer-reviewed and formally published. If this work was previously peer-reviewed and published, in the cover letter please provide the reason that this work does not constitute dual publication and should be included in the current manuscript.

4. We note that Figure 1 in your submission contain copyrighted images. All PLOS content is published under the Creative Commons Attribution License (CC BY 4.0), which means that the manuscript, images, and Supporting Information files will be freely available online, and any third party is permitted to access, download, copy, distribute, and use these materials in any way, even commercially, with proper attribution. For more information, see our copyright guidelines: http://journals.plos.org/plosone/s/licenses-and-copyright.

4.1.    You may seek permission from the original copyright holder of Figure 1 to publish the content specifically under the CC BY 4.0 license.

4.2.    If you are unable to obtain permission from the original copyright holder to publish these figures under the CC BY 4.0 license or if the copyright holder’s requirements are incompatible with the CC BY 4.0 license, please either i) remove the figure or ii) supply a replacement figure that complies with the CC BY 4.0 license. Please check copyright information on all replacement figures and update the figure caption with source information. If applicable, please specify in the figure caption text when a figure is similar but not identical to the original image and is therefore for illustrative purposes only.

Reviewers' comments:

Reviewer's Responses to Questions

**Comments to the Author**

1. Is the manuscript technically sound, and do the data support the conclusions?

Reviewer #1: Partly

Reviewer #2: Partly

Reviewer #3: Yes

2. Has the statistical analysis been performed appropriately and rigorously? 

Reviewer #1: No

Reviewer #2: No

Reviewer #3: N/A

3. Have the authors made all data underlying the findings in their manuscript fully available?

Reviewer #1: Yes

Reviewer #2: Yes

Reviewer #3: No

4. Is the manuscript presented in an intelligible fashion and written in standard English?

Reviewer #1: Yes

Reviewer #2: No

Reviewer #3: Yes

5. Review Comments to the Author

Reviewer #1: The theme is relevant and highlights the importance of multimorbidity as an important health indicator, especially in the rural population. However, the article needs to be better reformulated as suggested below.

1- Introduction: It is long and long-winded and does not address the target audience of the study. There are recent articles, including published in Plos Uma that addresses multimorbidity in a rural population that was not mentioned in this article. I suggest consulting the databases in a systematic way and restructure the introduction, addressing the context of the research problem, the literature gap and the objective / hypothesis you want to achieve. Following is the link to the manuscript that addresses the topic: https://doi.org/10.1371/journal.pone.0225416

2- Methodology: Would you like the authors to clarify better how the invitation for the research subjects was? How was an approach? Was there any criterion that these items actually portrayed the object of study? Was there any criterion of saturation of the characteristics or was this fact not adopted in the methodology? Describe whether the interview protocol has been previously tested.

3- Results: The results are very interesting and can be the best result, in the most descriptive and non-analytical style.

4- Conclusion: too long and repeating results. Highlight the contributions of the study and the social impact of the current scientific production..

Reviewer #2: This study examined multimorbidity and its effect on perceived burden, capacity and the ability to self-manage in a low-income rural primary care population.

Even it is a mall quantitative study the results may be useful. However this study has a number of major limitations that make the results difficult to interpret.

Therefore, I recommend accepting this paper and publishing it with some major revisions and after the polishing the English language.

Please find below specific comments.

1. Most studies do not use a hyphen for ‘multimorbidity’. To be consistent with literature, I would suggest to replace ‘multi-morbidity’ into ‘multimorbidity’.

2. Lines 62-64: the authors mentioned: More recently, there has been a recognition that many chronic conditions do not exist in isolation, but as clusters of conditions. Please add a reference.

3. Line 64: the authors mentioned: Although prevalence counts vary due to differences…. Please explain, give some examples.

4. Line 72: Please give the definition of multimorbidity which was used for your study.

5. Lines 88-89: the authors mentioned: The interaction is also subject to a social gradient. This is confusing for the reader. Please provide some explanations.

6. Lines 110-112: Authors mentioned that the aim of the study was to explore in a low-income rural population, how the patient experience of multimorbidity affects perceived capacity and workload. Any previous studies which examined that? Any literature for the background for that aim?

7. Line 130: the authors mentioned: “We were interested in exploring this in a population with common chronic health conditions...” Please add a reference.

8. Line 137: I suggest the authors to use just “study design” instead of “overall study design”.

9. Line 139: authors mentioned that they employed a phenomenological methodology. Please explain.

10. Lines 147-148: How did you obtain that information? How did you code and classify which are health condition? Did the authors used specific criteria for disease inclusion?

11. Line 157: Authors mentioned that participants provided their details with permission. Can you please clarify in the text the approval of the study by a Bioethics Committee as well as if the participants were required to sign a consent form and not why is the case?

12. In methods section please provide some details about the measures of the study (variables etc.)

13. Line 214: The phone interview due to COVID-19 restrictions need to be mentioned in the limitation section.

14. Please clarify why P6, P12 etc. are mentioned between the paragraphs. This is confusing for the reader. It could be mentioned in a table.

15. Lines 245-248: Split/re-arrange this sentence to make it easier for the reader to read.

16. Lines 252-253: authors mentioned that loss of physical capacity was strongly associated with perceived conditions importance. It is not specified how the authors find that results. Why strongly? Any test? This is a very crucial aspect for the analyses and thus it needs to be revised.

17. Line 265: Psychological capacity was also essential. How did the authors find this? Please add details.

18. Line 292: Increased healthcare….of income. How did the authors find that association? Is it an association or just an observation?

19. Lines 327-328: Please add a reference.

20. Lines 355-356: authors mentioned that “this could lead to depression…planning”. Please explain.

21. Lines 441-443: authors mentioned that lifestyle recommendations were often questioned because participants felt they were unrealistic or because people did not know how to undertake exercise when they had coexisting chronic pain. It needs to be mentioned in the limitation section.

22. Line 480: Multimorbidity was associated with significantly more treatment. Why significantly? Any test? This is a very crucial aspect for the analyses and thus it needs to be revised.

23. In Discussion section no comparison has been made with other studies. Why were those comparisons not included in the discussion?

Reviewer #3: 1. Abstract – clear and summarises the study appropriately.

2. Introduction – sets the context for the study. Provides rationale for undertaking the work. Conveys nuance in understanding and treating multimorbidity.

3. Line 93-106 - It was not clear to me how ‘workload’ within the CCM applies to the burden of multimorbidity – please clarify for the naïve reader.

4. It was not clear how the argument that disadvantaged populations are at increase risk of capacity-workload imbalance was reached. Please provide evidence for this assertion.

5. Line 110 - Explain further how ‘multimorbidity is an encounter with complexity’ and for whom.

6. Line 139 - You say you employed phenomenology – how so? Expand and revisit its utility in the discussion briefly.

7. Participant recruitment - You say the study focusses on low income rural primary care patients but your participant recruitment description does not make it clear that this was the population you recruited. Nor is this necessarily clear in the participant table, even if the ‘source of income’ suggests low income.

8. The findings are well-presented and it is clear how the framework analysis has been applied.

9. Line 140 – this phrasing suggests quantitative analyses were conducted – rephrase.

10. Line 110-128 I was asking the question where this study aims to rework Shippee’s orginal model and, if so, should this be done at the discussion stage based on data-driven induction? (e.g. application of Theory of Patient Capacity, NPT). Though I have noted you used framework analysis. Moving then onto the discussion, I would suggest you make your enhancement/reframing of the model in line with findings explicit in the discussion.

11. The discussion roots the findings in pre-existing literature.

6. PLOS authors have the option to publish the peer review history of their article (what does this mean?). If published, this will include your full peer review and any attached files.

Reviewer #1: No

Reviewer #2: No

Reviewer #3: **Yes: **Dr Maria Kordowicz

---

## [Author Response · Author response to Decision Letter 0]

6 Jun 2021

The full response to editor and reviewer comments is available in the attachment 'Response to Reviewers'.

---

## [Decision Letter · Decision Letter 1]

26 Jul 2021

Multi-morbidity and its effect on perceived burden, capacity and the ability to self-manage in a low-income rural primary care population: a qualitative study.

PONE-D-20-39711R1

Dear Dr. Hardman,

We’re pleased to inform you that your manuscript has been judged scientifically suitable for publication and will be formally accepted for publication once it meets all outstanding technical requirements.

Kind regards,

Filipe Prazeres, MD, MSc, Ph.D.

Academic Editor

PLOS ONE

Additional Editor Comments (optional):

Reviewers' comments:

Reviewer's Responses to Questions

**Comments to the Author**

1. If the authors have adequately addressed your comments raised in a previous round of review and you feel that this manuscript is now acceptable for publication, you may indicate that here to bypass the “Comments to the Author” section, enter your conflict of interest statement in the “Confidential to Editor” section, and submit your "Accept" recommendation.

Reviewer #2: All comments have been addressed

2. Is the manuscript technically sound, and do the data support the conclusions?

Reviewer #2: Yes

3. Has the statistical analysis been performed appropriately and rigorously? 

Reviewer #2: N/A

4. Have the authors made all data underlying the findings in their manuscript fully available?

Reviewer #2: Yes

5. Is the manuscript presented in an intelligible fashion and written in standard English?

Reviewer #2: Yes

6. Review Comments to the Author

Reviewer #2: (No Response)

7. PLOS authors have the option to publish the peer review history of their article (what does this mean?). If published, this will include your full peer review and any attached files.

Reviewer #2: No

---

## [Editor Report · Acceptance letter]

30 Jul 2021

PONE-D-20-39711R1 

Multimorbidity and its effect on perceived burden, capacity and the ability to self-manage in a low-income rural primary care population: a qualitative study. 

Dear Dr. Hardman:

I'm pleased to inform you that your manuscript has been deemed suitable for publication in PLOS ONE. Congratulations! Your manuscript is now with our production department. 

Kind regards, 

on behalf of

Prof. Filipe Prazeres 

Academic Editor

PLOS ONE